# A Fructan Exohydrolase from Maize Degrades Both Inulin and Levan and Co-Exists with 1-Kestotriose in Maize

**DOI:** 10.3390/ijms22105149

**Published:** 2021-05-13

**Authors:** Silin Wu, Steffen Greiner, Chongjian Ma, Jiaxin Zhong, Xiaojia Huang, Thomas Rausch, Hongbo Zhao

**Affiliations:** 1Key Laboratory of Biology and Genetic Improvement of Horticultural Crops, College of Horticulture, South China Agricultural University, Ministry of Agriculture and Rural Affairs, Guangzhou 510642, China; wusilin@stu.scau.edu.cn (S.W.); hxj2019@stu.scau.edu.cn (X.H.); 2Centre for Organismal Studies Heidelberg, Department of Plant Molecular Physiology, University of Heidelberg, 69120 Heidelberg, Germany; steffen.greiner@cos.uni-heidelberg.de (S.G.); jiaxin.zhong@cos.uni-heidelberg.de (J.Z.); thomas.rausch@cos.uni-heidelberg.de (T.R.); 3Department of Horticulture, Henry Fok College of Biology and Agricultural Science, Shaoguan University, Shaoguan 512005, China; ma_chj@hotmail.com

**Keywords:** *Zea mays*, fructan exohydrolase, inulin, levan, 1-kestotriose

## Abstract

Enzymes with fructan exohydrolase (FEH) activity are present not only in fructan-synthesizing species but also in non-fructan plants. This has led to speculation about their functions in non-fructan species. Here, a cell wall invertase-related Zm-6&1-FEH2 with no “classical” invertase motif was identified in maize. Following heterologous expression in *Pichia pastoris* and in *Nicotiana benthamiana* leaves, the enzyme activity of recombinant Zm-6&1-FEH2 displays substrate specificity with respect to inulin and levan. Subcellular localization showed Zm-6&1-FEH2 exclusively localized in the apoplast, and its expression profile was strongly dependent on plant development and in response to drought and abscisic acid. Furthermore, formation of 1-kestotriose, an oligofructan, was detected in vivo and in vitro and could be hydrolyzed by Zm-6&1-FEH2. In summary, these results support that Zm-6&1-FEH2 enzyme from maize can degrade both inulin-type and levan-type fructans, and the implications of the co-existence of Zm-6&1-FEH2 and 1-kestotriose are discussed.

## 1. Introduction

Fructans are fructose polymers. Fructan-synthesizing plant species represent about 15% of flowering plants, most conspicuously in the *Compositae, Liliaceae* and *Gramineae* families [1,2,3]. Fructans can be divided into three types: (1) an inulin type composed of β(2→1)-fructosyl linkage; (2) a levan type of fructan primarily linked by β(2→6)-linked fructosyl units; (3) the mixed type of fructan consisting of both β(2→6)- and β(2→1)-linked fructosyl units [4].

Fructan metabolism is dynamically controlled during plant development [5] and in response to various stress cues [6]. Oligofructans are oligosaccharides that exist obviously in plants such as onion, chicory, garlic and asparagus [7]. In most cases, oligofructans are mixtures of short-chain inulin-type fructans, namely 1-kestotriose (degree of polymerization (DP) equal to 3), nystose (DP4) and 1-fructofuranosylnystose (DP5) [8]. In addition to the role of fructans with high DP that serve as carbohydrate storage compounds, oligofructans are known to have membrane-protecting properties during cold adaptation [9,10]. Furthermore, as the smallest oligofructan with only two fructose units [11], 1-kestotriose may function as a signal molecule during plant growth and development [12].

Previous work on fructan-synthesizing plant species has revealed specific enzyme sets [13,14,15,16] for fructan biosynthesis and degradation, which relate to different fructan types [17,18,19]. In fructan plants, fructan is hydrolized by fructan exohydrolase (FEH) [20,21]. Based on substrate differences, FEHs are classified into three types: (1) 1-FEH mainly hydrolyzes inulin-type fructan; (2) 6-FEH mostly degrades levan-type fructan; (3) 6&1-FEHs degrade mixed types of fructan. FEH functions have been well researched in fructan plants during their growth and development [2,21,22,23]. The enzyme structure of FEH is closely related to the structure of cell wall invertase (CWI) [24,25]. The 3D structures of a CWI from *Arabidopsis* and an FEH from *Cichorium intybus* have been uncovered, which showed a clarification that there is a five-bladed β-propeller domain in an N-terminal as well as a C-terminal domain formed by two β-sheets to make up those three proteins [24,25]. This supports a hypothesis that FEHs may evolve from CWIs.

A first report in 2003 revealed the unexpected presence of fructan-degrading enzymes, i.e., fructan 6-exohydrolases (FEHs), in non-fructan plants [26]. Further studies confirmed that enzymes with FEH activity are encoded in the genomes of many non-fructan plant species [12,16]. However, their physiological roles have remained largely enigmatic. Recently, Huang et al. [27] reported the presence of a 6-FEH enzyme in maize, the expression of which was induced upon challenge with different bacteria bearing exo-polysaccharide coatings [28], however, a protective effect against these bacteria remains to be shown. Mechanistically, another possible role of FEH enzymes in non-fructan species may be to degrade 1-kestotriose and some related oligofructans [29,30]. In maize, previous studies have indicated that it may contain low amounts of fructans [31,32], but the presence of 1-kestotriose or related oligofructans has not yet been unequivocally confirmed, due to methodological constraints.

In the present study, a novel CWI-related fructan exohydrolase, Zm-6&1-FEH2, was cloned and characterized, which was able to hydrolyze both inulin-type and levan-type fructans. We also describe its subcellular localization and expression during maize growth and development and in response to drought and abscisic acid. In addition, oligofructans can indeed be detected in vivo and in vitro in different maize tissues, their amounts being affected by drought exposure, and an additional Zm-6&1-FEH2 is able to hydrolyze 1-kestotriose and oligofructans.

## 2. Results

### 2.1. Maize Cell-Wall Invertase-Related Enzyme Zm-6&1-FEH2 with Both Inulin-Type and Levan-Type Fructan Exohydrolase Activities

Previous work on maize cell-wall invertase homologs had identified four putative CWI isoforms, Zm-INCW1-4 [33]. Mining the GenBank database for as yet non-characterized CWI-related maize cDNAs, a candidate was found and named as Zm-6&1-FEH2 (Figure 1). The predicted protein sequence of Zm-6&1-FEH2 (Genebank No.: BT055913) displayed the β-fructosidase motif NDPNG/A and the canonical cysteine-containing catalytic site MWECP (Figure 1). In the phylogenic tree, Zm-6&1-FEH2 was grouped in a clade with several other monocot FEH enzymes (Figure 2a). As most of the FEHs characterized up to now have a low isoelectric point (pI), while typical cell wall invertases can have a high pI for improved binding to the cell wall [34], the pI of Zm-6&1-FEH2 was 6.06 (Figure 2a). In addition, regarding the Asp_239_XXLys_242_ motif, which was previously proposed to be a positive signature for CWI [24], Zm-6&1-FEH2 lacks the basic residue of this motif, confirming its distant relationship with CWI enzymes (Figure 2b–d). Furthermore, Zm-6&1-FEH2 has a high similarity with *Triticum aestivum* 1-FEH-w1, 1-FEH-w2 and *Lolium perenne* 1-FEH in this motif region. All these analyses indicate that Zm-6&1-FEH2 may have FEH activity but not a CWI (Figure 2b–d).

To identify the substrate specificity, Zm-6&1-FEH2 was expressed in *Pichia pastoris* strain X-33 (Figure 3). The result displayed that Zm-6&1-FEH2 had very low invertase activity and high activity towards inulin, levan and kestoses, i.e., being more active against fructans as compared to sucrose (Figure 4a), in accordance with the lack of Asp_239_XXLys_242_ CWI motif (see above). Thus, for consistency, we have assigned it as an FEH enzyme (with very low invertase activity). For comparison, we also expressed Zm-6&1-FEH2 transiently in leaves of *Nicotiana benthamiana*, and FEH activity towards sucrose, inulin and levan was determined in salt-eluted (i.e., cell wall-bound) protein fractions. To account for the presence of endogenous invertase and/or FEH activities from *Nicotiana benthamiana*, mock transformations with *Agrobacterium tuemfaciens* bearing the empty vector were performed to correct for this background. Again, the cell wall fractions revealed substantial enzyme activities with levan and inulin as substrates accompanying with very low activity with sucrose (Figure 4b). These results convincingly demonstrated that Zm-6&1-FEH2 is not a classic invertase, but a 6&1-FEH.

In contrast to the fructosyltransferases, many fructan exohydrolases are inhibited by sucrose at the protein level [35]. Further comparison of the GWAS and MLYTG motifs in FEHs, as it has been proved in the Ser 101 point of chicory 1-FEH IIa, showed very strong sucrose inhibition in the amino acid with S/G, with other amino acids such as Q/I/L/R showing very weak sucrose inhibition [34]. Further analysis of recombinant Zm-6&1-FEH2 showed no inhibitory effect of FEH activity with increased sucrose concentration (Figure 5).

### 2.2. Zm-6&1-FEH2 Localized to the Apoplast

Analysis of the full-length protein sequence by PSORT, Target P and SIGNAL P suggests that Zm-6&1-FEH2 protein is targeted to the apoplast (Appendix A). To verify the predicted apoplastic targeting, a construct coding for a maize FEH::GFP fusion protein was generated under the control of the cauliflower mosaic virus 35S promoter for stable transformation into *Arabidopsis*. Transformation with GFP alone displayed fluorescence in the nuclei and cytoplasm of *Arabidopsis* root cells (Figure 6b). By contrast, transformation with FEH::GFP fusion construct showed fluorescence was restricted to the apoplast of *Arabidopsis* root (Figure 6a). 

### 2.3. Zm-6&1-FEH2 Expressed Differently during Plant Development and in Response to Drought Stress

To verify the spatial and temporal pattern of expression, we performed real-time PCR using RNA isolated from leaf, root, stem, silk, anther and pollen, as well as from a range of early kernel development time points. Results of *Zm-6&1-FEH2* showed it expressed mainly in vegetative tissues (Figure 7a), and its mRNA level increased as leaves progressed from sink (basal part of leaf) to source (middle part of leaf) stages; other highly expressed tissues were root, silk and stem (Figure 7a). In seeds, the expression of *Zm-6&1-FEH2* was mainly restricted to the first week of pollination (Figure 7a).

To monitor the effects of drought and ABA stress on *Zm-6&1-FEH2* expression, transcript levels of *Zm-6&1-FEH2* were analyzed by real-time PCR along the maize leaf axis (one leaf was divided into 6 equal parts) and in several other maize tissues. The results showed that *Zm-6&1-FEH2* was up-regulated by drought, ABA and cold stress in the source (emerged) part of the leaf, but not in the sink (enclosed) part of the leaf (Figure 7b). In addition, drought up-regulated *Zm-6&1-FEH2* expression in root, stem, silk and the early stage of seed development (Figure 7c).

### 2.4. Zm-6&1-FEH2 Hydrolyze Oligofructan 1-Kestotriose from Maize

Previous work has reported the presence of fructans in maize, a non-fructan species [31,32]. However, since exclusively enzymatic methods were applied, i.e., fructose determination after fructanase treatment, the fructan profiles of different maize tissues were determined in this study by HPAEC-PAD analysis to identify individual oligofructans in different maize tissues. In maize pollen, 1-kestotriose was consistently detected, albeit at small amounts compared to sucrose (Figure 8). Occasionally, there were additional peaks co-chromatographed with 1,1-nystose and 1,1,1-kestopentaose, but the amounts were much lower than for 1-kestotriose (Figure 8). In other maize tissues like leaves, roots and stem, 1-kestotriose became detectable only after drought treatment, whereas exposure to abscisic acid (ABA) had only a minor effect (data not shown). In summary, these observations confirm that 1-kestotriose is indeed a metabolite in different maize tissues, and its abundance may increase in response to drought stress.

To clarify whether recombinant Zm-6&1-FEH2 protein was able to hydrolyze 1-kestotriose, the carbohydrate extracts from pollen, which were collected in 4 days after the pollination stage, were co-incubated with 25 µg of recombinant Zm-6&1-FEH2 at 30 °C for 1 h. Results clearly show the disappearance of the 1-kestotriose peak (identified by co-chromatography with authentic 1-kestotriose standard; Figure 8).

### 2.5. In Vitro Synthesis of Fructan Trisaccharides in Maize 

As no high-DP fructans can be found in maize tissues in vivo [31,32], the question arises as to whether fructan formation can be induced by another stimulus in vitro. Using the leaf disc of the source (middle) part of the fourth young leaf from 15 days after germination of a maize seedling, we treated them under 0, 5,10, 20 and 40% sucrose in the dark for 2 days’ incubation. The results demonstrate that sucrose feeding can induce 1-kestotriose production, and with increasing sucrose concentration higher amounts of 1-kestotriose are accumulated (Figure 9). In addition, 6-kestose and nystose appeared when sucrose concentration was added to 20–40% (Figure 9). Therefore, it is clearly demonstrated that maize leaves are capable of producing fructan trisaccharides such as 1-kestotriose, 6-kestose and neokestose from sucrose induction in the dark in vitro, but do not accumulate or produce higher DP fructans.

## 3. Discussion

Previous research on the protein structures of cell wall invertase and fructan exohydrolase has revealed the presence of N-terminal five-bladed β propeller domains with three highly conserved acidic motifs, MN**D**PNG, R**D**P and W**E**CP [34], which also appear in Zm-6&1-FEH2. Conversely, the “Asp/Lys” or “Asp/Arg” couples, which in the active site of cell wall invertase were shown to be important to stabilize the glucose moiety from sucrose [34], were absent in Zm-6&1-FEH2. Although these couples were considered as an important signature to distinguish FEH and cell wall invertase, some exceptions still appeared, for example, rice VI isoforms OsVIN1 and 2 with the “Asp/Lys” motif displayed substantial FEH activities [30]. Hence, enzyme characterization of Zm-6&1-FEH2 was necessary. The results obtained via heterologous expression of Zm-6&1-FEH2 in *Pichia pastoris* and transient expression in *Nicotiana benthamiana* showed it had FEH activity rather than invertase activity (Figure 4). Together, these observations confirm that Zm-6&1-FEH2 belongs to family 32 of the glycoside hydrolases (GH32), which groups together with cell-wall-type invertases.

Previous work has revealed the presence of fructan exohydrolases in several non-fructan plant species, dicots and monocots, sometimes considered “defective invertases” with 6&1-FEH side activities [4]; however, there is still debate about their physiological roles. According to a current hypothesis, a 6-FEH in maize is thought to be involved in defense against bacterial pathogens with exo-polysaccharide coatings [27]. However, as small amounts of endogenous 1-kestotriose have been detected in *Arabidopsis thaliana* [4] it cannot be excluded that 6&1-FEH also have a role in regulating endogenous 1-kestotriose levels in non-fructan plants. This study demonstrates for the first time the presence of endogenous 1-kestotriose and Zm-6&1-FEH2 in a monocot non-fructan crop species.

The presence of the short oligo-fructan 1-kestotriose in maize pollen was demonstrated by HPAEC-PAD-based fructan profiling (Figure 8) and its degradability by incubation with recombinant Zm-6&1-FEH2 enzyme (Figure 8). The observation of a drought-induced increase of endogenous 1-kestotriose levels in leaves (data not shown) suggests that environmental stress may increase 1-kestotriose levels. In general, fructan synthesis in plants requires the presence of fructosyltransferase enzymes. Most plants with FEH enzymes (including non-fructan plants) also have fructan synthesis and accumulation ability [12,20,30,36,37]. This work speculated maize may lack fructosyltransferases for fructan chain elongation. After systematically blast searching in NCBI, MaizeGDB (Maize Genetics and Genomics Datebase), Maize Sequence and Query Sequence Visualizer databases, results show that maize indeed lacks fructosyltransferase genes, and only two maize vacuolar invertases (Ivr1 and Ivr2) were found as fructosyltransferase homologues. Interestingly, vacuolar invertase and fructosyltransferase all use sucrose as the substrate, and they are very similar at the biochemical level. We conclude that the evolutionarily related vacuolar invertase(s) is the most likely origin of 1-kestotriose formation in non-fructan plants [4,29,30]; it may be relevant that numerous studies have demonstrated a drought-induced increase of VI expression [33,38]. While such a scenario assumes that 1-kestotriose formation occurs in the vacuole, the transport to the apoplast via vesicle transfer to bring 1-kestotriose into direct contact with apoplastic Zm-6&1-FEH2 enzyme cannot be excluded [39].

Previously, oligofructans were hypothesized to serve as signal molecules (or elicitors) in plants, acting at very low concentration (nM range) [4]. In line with this assumption, our results suggest that Zm-6&1-FEH2 may control 1-kestotriose levels in vivo. Thus 1-kestotriose-mediated signaling may be involved in stress responses, acting as endogenous, phloem-mobile stress signals between source and sink tissues. Furthermore, sucrose metabolism and signaling may possibly be linked to oligofructan metabolism, and sugar signaling may relate the hexokinase-dependent pathway to the regulation of FEH activity. This could result in fine-tuning oligofructans signaling under different stresses in maize. That is, FEHs might be involved in removing the oligofructans (especially 1-kestotriose) signals, and even be involved in the process of sensing these signals. Finally, 1-kestotriose may help to increase maize tolerance to adverse conditions, like cold and drought. Former reports have indicated that endogenous oligofructans (especially 1-kestotriose and kestopentaose) increased significantly under low temperature and high C_2_O treatment, which could provide protection against damage caused by storage at suboptimal low temperatures in table grapes [40]. In addition, maize oligofructans may induce stomatal closure like ABA to resist drought. It has been reported that burdock oligofructans can induce stomatal closure in *Pisum sativumis* mediated by reactive oxygen species (ROS) and ROS-dependent nitric oxide (NO) production [41]. Interestingly, the activity of FEHs under abiotic stress increases in many studies, but whether there is a link between them and oligofructans is unclear. Last, defective invertase Nin88 was found to play a crucial role in the early stages of pollen development in tobacco [4], but whether Zm-6&1-FEH2 also functions as a defective invertase still needs consideration.

## 4. Materials and Methods

### 4.1. Plant Material and Cultivation

*Nicotiana benthamiana* L. and *Zea mays* L. (SEVERUS, KWS) were planted in a greenhouse at 25 ± 2 °C, with a 16 h light period with 300 µmol m^−2^ s^−1^. Plant tissue samples were either directly used or immediately frozen in liquid nitrogen and stored at −80 °C until use.

### 4.2. RNA Extraction, Cloning, Sequencing and Phylogeny

Total RNA was extracted with the GeneMATRIX Universal RNA purification Kit (Roboklon, Berlin, Germany). cDNA synthesis was performed immediately after DNase (AppliChem, Darmstadt, Germany) treatment, using AMV-Reverse Transcriptase (Roboklon, Berlin, Germany). Full-length cDNAs of Zm-6&1-FEH2 were cloned from maize leaf cDNA by PCR (35 cycles: 95 °C/30 s–55 °C/30 s–72 °C/1 min/1 kb; final extension: 10 min), using Phusion High-Fidelity DNA polymerase with a GC buffer (Finnzymes, Schwerte, Germany) and corresponding primers presented in Appendix A. PCR products were fully sequenced (Starseq, Heidelberg, Germany) and cloned into pDONR201 or pDONRzeo vector (Invitrogen) to obtain the entry clone for the Gateway system (Invitrogen, Darmstadt, Germany).

Phylogenetic and molecular evolutionary analysis were conducted with clustal Omega [42] and MEGA 7 [43]. Accession numbers of genes used for alignment and phylogenetic trees were AF050129 (*Zea mays* INCW1), AF050631 (*Zea mays* INCW2), AF043346 (*Zea mays* INCW3), AP004156 (*Oryza sativa* CIN1), AL662945 (*Oryza sativa* CIN2), AL662945 (*Oryza sativa* CIN3), NM_112232 (*Arabidopsis thaliana* CWINV1), NM_115120 (*Arabidopsis thaliana* CWINV2), NM_104385 (*Arabidopsis thaliana* 6-FEH), NM_129177 (*Arabidopsis thaliana* CWINV4), NM_121230 (*Arabidopsis thaliana* 6&1-FEH), AJ534447 (*Hordeum vulgare* CWINV1), Z35162 (*Vicia faba* CWINV1), M58362 (*Daucus carota* CWINV1), X78424 (*Daucus carota* CWINV2), Z22645 (*Solanum tuberosum* CWINV1), Q9M4K7 (*Solanum tuberosum* CWINV2), Q9M4K8(*Solanum tuberosum* CWINV3), X81834 (*Nicotiana tabacum* CWINV1), AF376773 (*Nicotiana tabacum* CWINV2), AJ272304 (*Lycopersicum esculentum* CWINV1), AB004558 (*Lycopersicum esculentum* CWINV2), Z35162 (*Vicia faba* CWINV1), AJ508534 (*Beta vulgaris* 6-FEH), AJ242538 (*Cichorium intybus* FEH-I), AJ295034 (*Cichorium intybus* FEH-IIb), AJ295033 (*Cichorium intybus* FEH-IIa), AJ509808 (*Campanula rapunculoides* 1-FEH), AF030420 (*Triticum aestivum* CWINV1), AJ516025 (*Triticum aestivum* 1-FEH-w1), AJ508387 (*Triticum aestivum* 1-FEH-w2), AM075205 (*Triticum aestivum* 6-FEH), AB089269 (*Triticum aestivum* 6&1-FEH), AB089271 (*Triticum aestivum* 6-KEH-w1), AB089270 (*Triticum aestivum* 6-KEH-w2), AJ605333 (*Hordeum vulgare* 1-FEH), DQ016297 (*Lolium perenne* 1-FEHa), AB583555 (*Phleum pratense* 6-FEH-1), EU971090 (*Zea mays* 6&1-FEH1), EU957945 (*Zea mays* 6-FEH), AF276703 (*Oryza sativa*VIN1), AF276704 (*Oryza sativa*VIN2).

### 4.3. Gene Expression Analysis by qPCR

qPCR analysis was performed with the Rotor-Gene Q system (Qiagen, Hilden, Germany) by using SYBR Green (S7563, Invitrogen, Darmstadt, Germany) to monitor dsDNA synthesis. Thermal cycling conditions were identical for all primer pairs: 95 °C/6min, followed by 40 cycles of 95 °C/20 s–58 °C/20 s–72 °C/20 s, followed by a melt cycle from 50 to 95 °C. To determine primer efficiency, serial dilutions of the templates were conducted for all primer combinations. Each reaction was performed in triplicate, and the amplification products were examined by agarose gel electrophoresis and melting curve analysis. The expression stability of reference genes (*ubiquitin* and *tubulin*) was assessed by using GeNorm algorithms, and the relative gene expression level was calculated by normalizing to the geometric mean of the reference genes according to a previously described method [44]. Primers for reference genes and target genes are presented in Appendix A. For each tissue, three independent cDNA preparations were analyzed with three technical replicates each.

### 4.4. Expression of Recombinant FEH Protein in Pichia Pastoris

The PCR amplified coding region of *Zm-6&1-FEH2* (primers in Appendix A) and pPICZαA vector (Invitrogen, Darmstadt, Germany) were digested with EcoRI and XbaI (New England Biolabs). DNA fragments were purified by using the NucleoSpin Extract II Kit (Macherey-Nagel) according to the manufacturer’s instructions. The purified product was ligated by using T4 DNA ligase (New England Biolabs, UK), with incubation at 14 °C for 16 h. The ligation product was transformed into *E. coli* competent DH5α cells by electroporation. Subsequently, bacterial cells were plated on a low-salt LB medium supplemented with zeocin as a selection marker. Positive colonies were used for vector amplification. pPICZαA plasmid with Zm-6&1-FEH2 (and empty vector as a control) was linearized by *Pme*I, and then transformed into *Pichia pastoris* strain X-33 via electroporation. Further selection and protein purification were performed as described [27].

### 4.5. Plant Transformation and Protein Extraction

The coding region of *Zm-6&1-FEH2* was cloned into the pB7WG2 vector downstream of the 35S promoter (primers in Appendix A) and was transformed into *Escherichia coli* competent DH5α cells by electroporation. Then this plasmid was transformed into *Agrobacterium tumefaciens* strain C58C1 cells by electroporation. Transient expression in 8 to 12 week *Nicotiana benthamiana* leaves was performed by *Agrobacterium* leaf infiltration [27], transformation with P19 served as a control to account for *Agrobacterium* transformation-induced induction of endogenous CWI and FEH activities. Extraction of soluble and cell-wall-bound proteins from *Nicotiana benthamiana* leaves and different maize tissues essentially followed the protocol [27]. Protein concentrations were quantified by Bradford assay (Roti^®®^-Quant; Roth).

### 4.6. Determination of FEH Activity

FEH protein was aliquot and was incubated with 6% (*w*/*v*) inulin (Sigma-Aldrich, Darmstadt, Germany), 1 mM levan (Sigma-Aldrich, Darmstadt, Germany) or 1–100 mM sucrose (Applichem, Darmstadt, Germany) in 50 mM NaOAc buffer, pH 5.0 at 37 °C for different time intervals. After incubation, the reaction was stopped by heating at 95 °C for 5 min. Released fructose was determined by HPAEC-PAD as described [27,45]. In parallel, glucose and fructose were also determined by a coupled spectrophotometric enzyme assay [46]. All enzyme measurements were performed under conditions where activities were proportional to enzyme amounts and incubation times.

### 4.7. Carbohydrate Extraction and Analysis

Total soluble carbohydrates were extracted in maize tissues [47]. In brief, water soluble carbohydrates were extracted from 200 mg frozen, homogenized tissue by incubation in 600 μL extraction buffer (50 mM sodium acetate pH 5, 10 mM NaHSO_3_, 0.1% Polyclar AT) for 15 min at 95 °C. After two centrifugations for 5 min at 10,000 g, the supernatants were dried in a speedvac concentrator (Bachofer, Reutlingen, Germany). Then, the sugar pellets were dissolved in HPLC-water (VWR Prolabo) to 10 mg/mL prior to analysis.

Carbohydrate quantification was performed by high-performance anion exchange chromatography (HPAEC) to determine glucose, fructose, sucrose, 1-kestotriose, 1,1-kestotetraose, 1,1,1-kestopentaose and recording of profiles for inulin and levan. Measurement and analysis were performed on a DionexICS-3000 system with an Electrochemical cell, SP-1 Single Pump, GM-4 gradient mixer, AS50 autosampler, Carbopac PA1 4 × 50 mm Guard column and Carbopac PA1 4 × 250 mm analytical column and operated with the Chromeleon 7.0 software (all components from Dionex). For eluent preparation (eluent A: 150 mM NaOH; eluent B: 150 mM NaOH, 700 mM sodium acetate) appropriate amounts of Chromanorm HPLC water (VWR Prolabo) were weighed in and sodium acetate was added into the eluent B. Then the eluents were degassed for 5 min by gassing with helium and supplemented with NaOH. Before the analytical run, the system was equilibrated by an initial equilibration run with the following settings: flow rate 1 mL/min, 0 min–10 min 3% A, linear gradient to 97% A at 12 min, 12 min–55 min 97% A. Carbohydrate quantification was achieved through co-chromatography of external standards with the following settings: flow rate 1 mL/min, 0 min–3 min 97% A, linear gradient to 71% A at 8 min, linear gradient to 3% A at 13 min. For inulin profiles a modified gradient was used: flow rate 1 mL/min, 0 min–3 min 97% A, linear gradient to 71% A at 8 min, linear gradient to 29% A at 50 min, linear gradient to 3% A at 55 min. For peak identification, glucose (Merck, Darmstadt, Germany), fructose (Sigma-Aldrich, Darmstadt, Germany), sucrose (Applichem, Darmstadt, Germany), 1-kestotriose, 1,1-kestotetraose and 1,1,1-kestopentaose (all Wako Chemicals, Neuss, Germany) were used as standards.

### 4.8. CLSM Analysis

Full-length cDNA of Zm-6&1-FEH2 without a stop codon was cloned into the pB7YWG2 vector (primers in Appendix A) and was then transformed into *E. coli* competent DH5α cells and into *Agrobacterium tumefaciens* strain C58C1 cells by electroporation. Microscopic analyses were carried out by using a confocal laser scanning microscope (LSM510 Meta, Zeiss, Jena, Germany). The following excitation and detection wavelength were used: GFP, excitation at 488 nm, detection at bandpass 505–530 nm; RFP, excitation at 543 nm, detection at bandpass 560–615 nm; YFP, excitation at 514 nm, detection at bandpass 530–560 nm. Chlorophyll autofluorescence: excitation at 488 nm, detection at longpass 650 nm.

### 4.9. Statistical Analysis

All gene expression tests and enzyme activity assays were performed in 3–4 independent experiments, with at least 3 technical replicates for each experiment. For further details see figure legends.

## Figures and Tables

**Figure 1 ijms-22-05149-f001:**
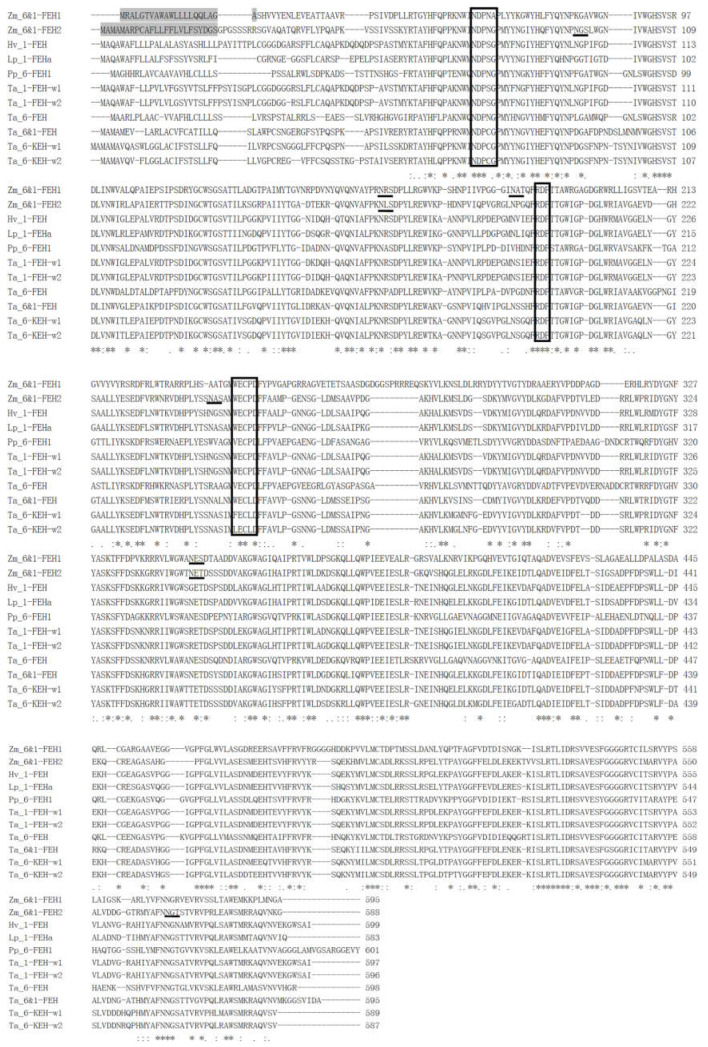
Amino acid sequence alignment of Zm-6&1-FEH2 and *Poaceae* species FEHs. β-fructosidase motifs (NDPNG/A), cysteine-containing catalytic sites (MWECP/V) and conserved Asp residues (D) are boxed. Putative glycosylation sites are underlined. The predicted N-terminal signal peptides are shaded. Species abbreviations are Hv, *Hordeum vulgare*; Lp, *Lolium perenne*; Pp, *Phleum pratense*; Ta, *Triticum aestivum*; Zm, *Zea mays*.

**Figure 2 ijms-22-05149-f002:**
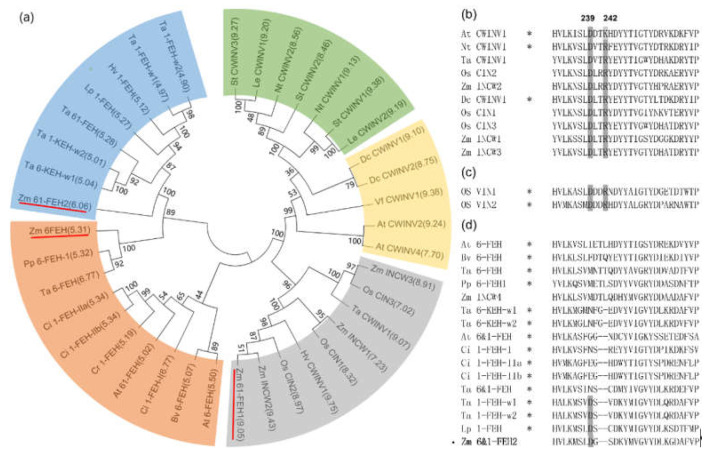
Zm-6&1-FEH2 is closely related to FEH, but not cell wall invertase. (**a**) Phylogenetic tree of FEHs and cell wall invertase-like cDNA-derived amino acid sequences. Groups grey, green and yellow are classified as CWI: CWIs from monocot species are classified in group grey, from dicot species in groups green and yellow. FEHs are classified in groups orange and blue: FEHs from dicot species are classified in group orange and those from monocot species in group blue. Zm-6&1-FEH2 is underlined. Isoelectric points are presented in brackets. (**b**–**d**) Multiple sequence alignment of the Asp-239XXLys-242 motif in CWIs, two *Oryza sativa* VIs and FEHs. Functionally characterized enzymes are marked with an asterisk in (**b**–**d**). Species abbreviations are At, *Arabidopsis thaliana*; Bv, *Beta vulgaris*; Ci, *Cichorium intybus*; Cr, *Campanula rapunculoides*; Dc, *Daucus carota*; Nt, *Nicotiana tabacum*; Hv, *Hordeum vulgare;* Le, *Lycopersicum esculentum*; Lp, *Lolium perenne;* Os, *Oryza sativa*; St, *Solanum tuberosum*; Pp, *Phleum pratense*; Ta, *Triticum aestivum*; Vf, *Vicia faba*; Zm, *Zea mays*.

**Figure 3 ijms-22-05149-f003:**
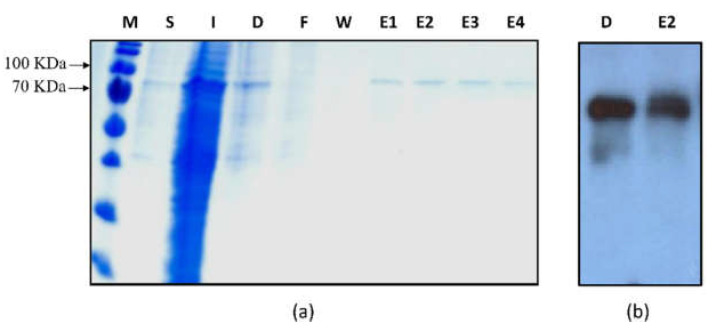
Heterologous expression of Zm-6&1-FEH2 in *Pichia pastoris*. (**a**) SDS-PAGE analysis of recombinant Zm-6-FEH protein. M, molecular weight marker; S, soluble proteins; I, insoluble proteins; D, dialyzed soluble proteins; F, column flow-through; W, column wash; E1-4, protein elutions. (**b**) Immunoblot analysis of dialyzed and eluted Zm-6&1-FEH2 proteins, which showed an immunoblot signal at about 80 kDa as detected with the C-myc antiserum.

**Figure 4 ijms-22-05149-f004:**
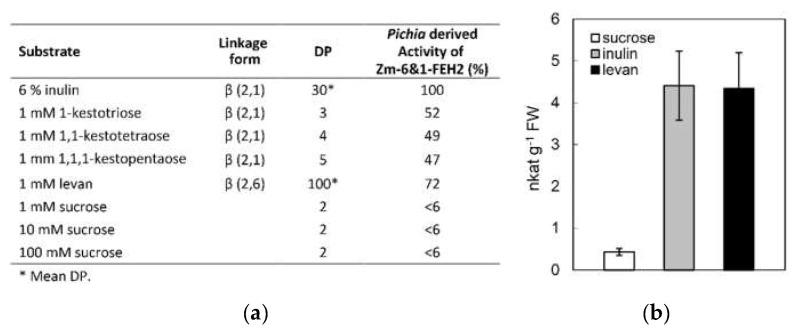
(**a**) Substrate specificity of recombinant maize Zm-6&1-FEH2 expressed in *Pichia pastoris*. Enzyme activity was determined from released fructose as quantified by HPAEC-PAD analysis. DP, fructan polymerization degree. (**b**) Cell-wall-associated invertase and fructan exohydrolase activities in *Nicotiana benthamiana* leaves transiently transformed with *Zm-6&1-FEH2*. Forty-eight hours after *Agrobacterium-tumefaciens*-mediated transformation of *Nicotiana benthamiana* leaves, enzyme activities were determined in salt-eluted (1M NaCl) cell wall protein fractions, collected on a Millipore filter with a cut-off of 50 kD. Substrate concentrations were 100 mM sucrose, 1 mM levan and 6% (*w*/*w*) inulin, respectively. Cell wall invertase activity induced by transformation with empty vector alone was subtracted.

**Figure 5 ijms-22-05149-f005:**
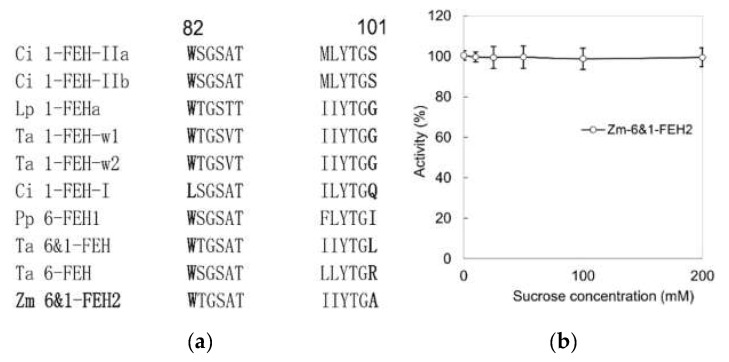
Activity of Zm-6&1-FEH2 is not inhibited by sucrose. (**a**) The GWAS and MLYTG motifs of FEHs in different plant species. (**b**) Zm-6&1-FEH2 activity under increased sucrose concentration. Activity is shown as a percentage of the maximum activity calculated by the fructose released after 25 µg of recombinant Zm-6&1-FEH2 co-incubated with 1 mM levan (mean DP 100). Data are means of 3 replicates.

**Figure 6 ijms-22-05149-f006:**
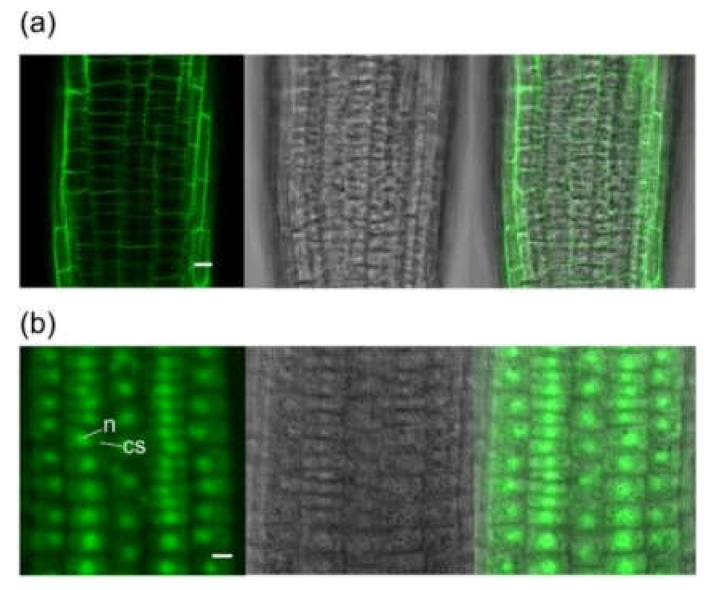
Zm-6&1-FEH2 is targeted to the apoplast. (**a**) Stable expression of Zm-6&1-FEH2::GFP fusion protein in root cells of *Arabidopsis*. (**b**) Stable expression of GFP alone in root cells of *Arabidopsis*. n: nucleus; cs: cytoplasm. Left: fluorescence signal from GFP; middle: bright field; right: merged GFP signals in bright field. Scale bar = 10 µm.

**Figure 7 ijms-22-05149-f007:**
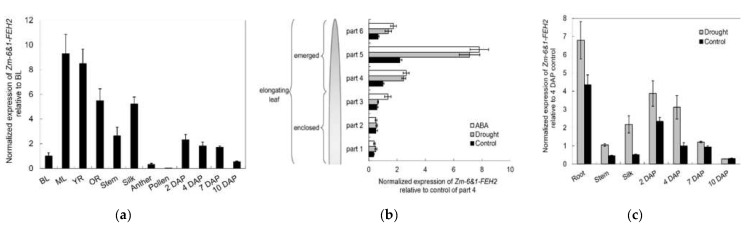
Expression profiles of Zm-6&1-FEH2. (**a**) Expression at transcript level for *Zm-6&1-FEH2* in different maize tissues. Expression data were normalized with respect to actin and ubiquitin, and are presented relative to the basal part of the leaf (BL). (**b**,**c**) Effect of drought and abscisic acid (ABA) treatments on the expression of *Zm-6&1-FEH2*. Expression data are presented relative to part 4 from the leaves of the control plants to seeds 10 DAP. (**b**) Expression profiles along the axis of the fourth leaf of 3-week-old greenhouse-cultivated maize seedlings had been determined under different growth conditions. After identical pre-cultivation for 12 days, subsequent treatments were as follows: control, regular watering at 3-day intervals; drought exposure, no additional watering during 9 days; abscisic acid (ABA) treatment, regular watering as in controls but supplemented with 50 µM ABA. Length of enclosed leaf parts (1–3): 3cm; length of emerged leaf parts (4–6): 6 cm. (**b**) Expression data from fully grown plants at/after pollination. Control, regular watering at 3-day intervals; drought, watering was stopped 7 days before pollination. BL, basal part of leaf; ML, middle part of leaf; YR, young root; OR, old root; DAP, days after pollination (seeds). BL, ML and YR refer to 3-week-old greenhouse-cultivated seedlings (at the 4-leaf-stage), BL and ML refer to the fourth leaf. All other samples were from fully grown plants at/after pollination. Results are means of 3 biological replicates (±SE), each with 4 technical replicates.

**Figure 8 ijms-22-05149-f008:**
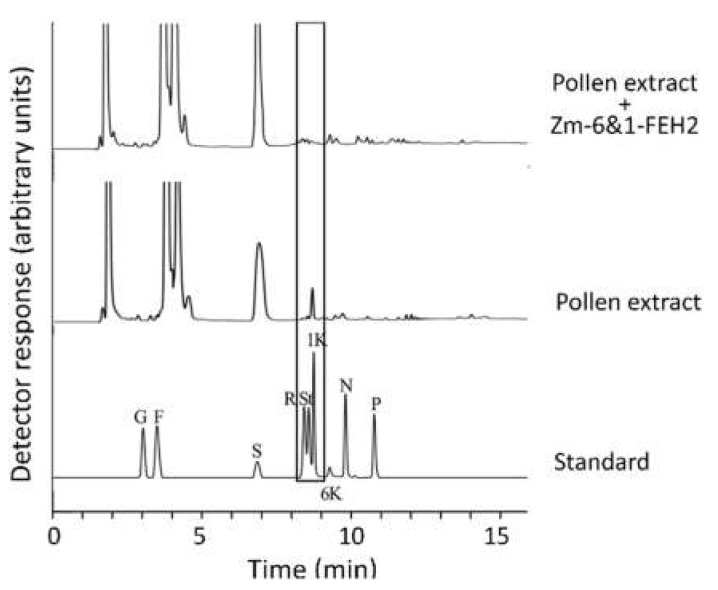
HPAEC-PAD chromatograms of incubation recombinant Zm-6&1-FEH2 with endogenous maize pollen fructan (1-kestotriose) after 1 h incubation. Abbreviations for each sugar peak in the standard are G, glucose; F, fructose; S, sucrose; R, raffinose; St, stachyose; 1K, 1-kestotriose; 6K, 6-kestotriose; N, 1,1-nystose; P, 1,1,1-kestopentaose.

**Figure 9 ijms-22-05149-f009:**
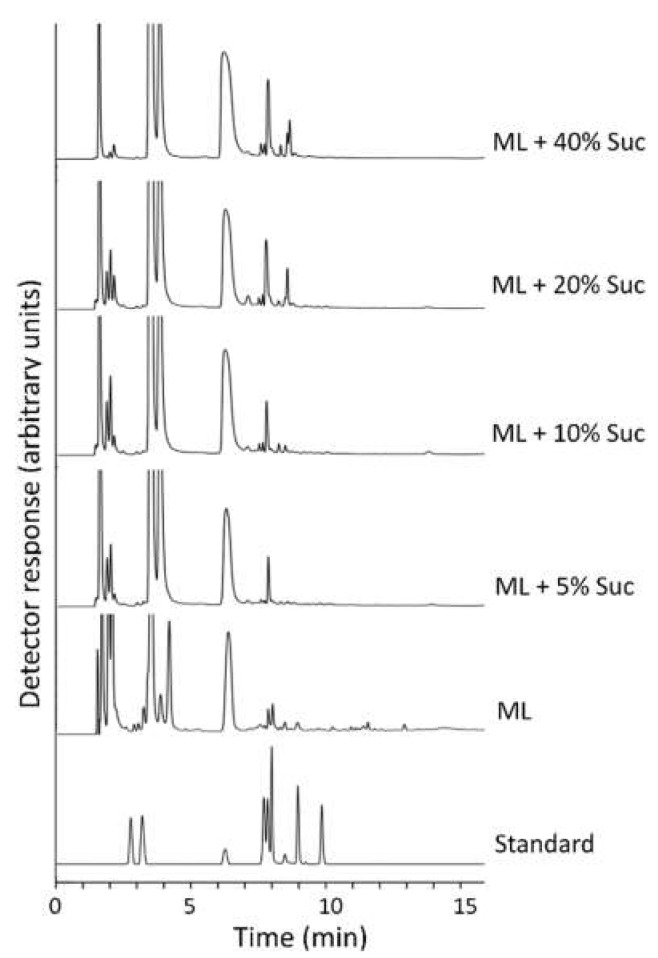
In vitro synthesis of fructan trisaccharides. HPAEC-PAD chromatograms of incubation of excised maize leaves with different sucrose concentrations in the dark. ML, middle part of leaf; Suc, sucrose.

## Data Availability

The data presented in this study are available on request from the corresponding author.

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
