# Peer review of "A Fructan Exohydrolase from Maize Degrades Both Inulin and Levan and Co-Exists with 1-Kestotriose in Maize"

_ijms, 2021, doi:10.3390/ijms22105149_

Round 1

Reviewer 1 Report

General comments to manuscript IJMS 1199746 

This manuscript entitled “ A fructan exohydrolase from maize degrades both inulin and 2 levan and co-exist with 1-kestotriose in maize “ by Wu. et al describe the production and expression of  fructan exohydrolase. The specific comments below should help the authors to improve the quality of the manuscript.

Specific comments:

Figure 3a : Lane D?

Figure 3: Why abbreviations of sugar were given in this legend of figure?

Figure 6: Identified the 3 parts of figure 6a and 3 parts of figure 6b.

Figure 9: Identified the other signals in the beginning of the chromatogram?

Line 383: Give more details of the eluent gradient used for the HPAEC analysis.

Reviewer 2 Report

Manuscript number: 1199746

Title: A fructan exohydrolase from maize degrades both inulin and 2 levan and co-exist with 1-kestotriose in maize

I like the effort of the authors to identify a fructan exohydrolase in maize and characterize this enzyme, its localization and expression during growth and different conditions. In addition, the authors detected the presence of oligofructans in maize tissues.

In the following paragraphs, I will provide clear information in order to improve the quality of the manuscript. In general, the English level is good, but I have detected several mistakes along the document. I would recommend the authors to revise carefully the manuscript.

ABSTRACT

In my opinion, the abstract summarizes correctly the manuscript.

  • Line 21: Furthermore, an oligofructan, 1-kestotriose, was found can be synthesized in vivo and in vitro in maize and hydrolyzed by Zm-6&1-FEH2. Please, rephrase.

INTRODUCTION

In general, the introduction is clear, fluid and gives the necessary context for the article. The objectives have been also exposed clearly. In my opinion, this section has been correctly performed. I have detected minor mistakes.

  • Line 61: “however, a protective effect against …”. Please, correct.
  • Line 67: “The present study cloned and characterized a novel CWI-related enzyme, Zm-6&1-67 FEH2, is able to hydrolyze inulin and levan type fructans.” Please, rephrase.

RESULTS

In my opinion, the authors have presented their results in a quite clear and precise way. Figures are visual and easy to understand. However, I would recommend the authors to revise carefully this section, since I have detected several phrases that should be rephrased (they are confusing). Some comments are explained below:

  • In Figure 2, it is not necessary to place the color of the groups between parentheses.
  • In Figure 3, the abbreviation of D is missing. In addition, sugar peaks that do not appear in the image are mentioned in the figure caption (Abbreviations for each sugar peak in the standard are: G, glucose; F, fructose; S, 134 sucrose; R, raffinose; St, stachyose; 1K, 1-kestotriose; 6K, 6-kestotriose; N, 1,1-nystose; P, 1,1,1-135 kestopentaose.)
  • In Figure 4, Pichia pastoris should be in italics.
  • Line 196-197: To further clarify the spatial and temporal pattern of drought, ABA and cold stress 196 effected Zm-6&1-FEH2 expression in maize leaf. One leaf was divided into… Please, rephrase.
  • I would suggest to move paragraphs 196-202 to Line 180.
  • Line 198: temporal pattern and stress response were also analyzed by… Please, correct.
  • Line 213: In maize pollen 1-kestotriose was consistently detected, albeit at 213 small amounts as compared to sucrose. Please, correct.
  • Line 221: To clarify should recombinant Zm-6&1-FEH2 can hydrolyze. Please, rephrase.
  • In my opinion, Figure 8 should be moved to the bottom of the section. Similar comment for Figure 9.
  • Line 231: one question is: can fructan be synthesized and accumulated by another stimulus in vitro? Please, correct.
  • Line 235-236: Results showed sucrose feeding can induce 1-kestotriose producing, and the higher 1-kestotriose accumulation followed by the increased sucrose concentration treatment. Please, rephrase.
  • Line 239: producing fructan trisaccharides. Please, correct.

DISCUSSION

The authors have discussed correctly their results, comparing the finding with previous studies. However, like the previous sections, I would recommend checking the writing.

MATERIAL AND METHODS

From my point of view, the scientific design and methodology employed is suitable for achieving the objectives proposed. However, I would recommend the authors to check the correct use of units (E.g.: 30sec, 1min à 30 seg, 1 min, different abbreviations: sec/s).

  • In line 319, there is a reference that does not have the same format as the others: (Larkin et al., 2007).
  • Line 366: Please, include the full name of Escherichia coli, since it is the first time mentioned in the manuscript.
  • Line 383: How the total carbohydrates were extracted. I would recommend including a brief description.

FINAL CONCLUSIONS

In my opinion, the authors have performed extensive work, with correct scientific bases and which provides interesting results. Therefore, I am suggesting MAJOR REVISIONS to improve its quality before publishing.

Round 2

Reviewer 2 Report

Manuscript number: 1199746

Title: A fructan exohydrolase from maize degrades both inulin and 2 levan and co-exist with 1-kestotriose in maize

The authors have considered all the comments suggested and they have carried out several changes to improve the quality of the manuscript. Regarding the English level and writing (the main limitation of the article), the authors have improved it. However, some minor changes are still necessary.

Introduction

Line 92: frucan. Please, correct.

Results and discussion

Line 510: As no high DP fructans can be found in maize tissues in vivo, consider of previous 510 paper showed maize can synthesis a huge amount of fructans [31,32] … Please, rephrase.

FINAL REMARKS

In my opinion, the authors have successfully increased the quality of their manuscript and it may be considered for publication.

Author Response

Introduction

Line 92: frucan. Please, correct.

Revised.

Results and discussion

Line 510: As no high DP fructans can be found in maize tissues in vivo, consider of previous paper showed maize can synthesis a huge amount of fructans [31,32] … Please, rephrase.

Revised.

As no high DP fructans can be found in maize tissues in vivo [31,32], the question arises whether fructan formation can be induced by another stimulus in vitro?